# Agreement between 2017 ACC/AHA Hypertension Clinical Practice Guidelines and Seventh Report of the Joint National Committee Guidelines to Estimate Prevalence of Postmenopausal Hypertension in a Rural Area of Bangladesh: A Cross Sectional Study

**DOI:** 10.3390/medicina55070315

**Published:** 2019-06-26

**Authors:** Lingkan Barua, Mithila Faruque, Palash Chandra Banik, Liaquat Ali

**Affiliations:** 1Department of Noncommunicable Diseases, Bangladesh University of Health Sciences, 125/1 Darus Salam, Mirpur-1, Dhaka 1216, Bangladesh; 2Department of Biochemistry and Cell Biology, Bangladesh University of Health Sciences, 125/1 Darus Salam, Mirpur-1, Dhaka 1216, Bangladesh

**Keywords:** postmenopausal women, hypertension, Bangladesh, American College of Cardiology/American Heart Association, Joint National Committee, agreement

## Abstract

*Background and objectives:* Justification for application of 2017 American College of Cardiology/American Heart Association (ACC/AHA) guidelines to detect hypertension (HTN) among Bangladeshi population is understudied. This prompted us to examine the level of agreement between 2017 ACC/AHA and Joint National Committee 7 (JNC 7) guidelines to detect postmenopausal HTN in a rural area of Bangladesh. *Materials and Methods:* This cross-sectional study recruited 265 postmenopausal women of 40–70 years of age who visited a rural primary health care centre of Bangladesh. HTN was diagnosed based on two definitions: the JNC 7 guidelines (SBP ≥ 140 or DBP ≥ 90 mmHg), and the 2017 ACC/AHA guidelines (SBP ≥ 130 mmHg, or DBP ≥ 80 mmHg). The prevalence of postmenopausal HTN, its sub-types and stages were reported and compared using frequency and percentage. Agreement was evaluated using Cohen’s Kappa (κ), Prevalence-Adjusted Bias-Adjusted Kappa (PABAK) and First-order Agreement Coefficient (AC1). *Results:* The prevalence of postmenopausal HTN was 67.5% and 41.9% using 2017 ACC/AHA and JNC 7 guidelines respectively. Among the HTN sub-types and stages, the new 2017 ACC/AHA guideline classified higher proportion of respondents as having isolated systolic hypertension (ISH) (42.6%) and stage 2 HTN (35.8%) compared to JNC 7 (28.7% and 6.8% respectively). On the other hand, the JNC 7 guideline identified more respondents as pre-hypertensive (32.5%) when compared with the 2017 ACC/AHA guideline (3.8%). Between two guidelines, highest agreement was observed for ISH (86.03%) and those had pre-hypertension/elevated blood pressure (71.3%). Similarly, Landis & Koch’s approach detected highest agreement for ISH (κ = 0.74, substantial; PABAK = 0.76, substantial; AC1 = 0.84, excellent; *p* < 0.001) and pre-hypertension/elevated blood pressure (κ = 0.12, slight; PABAK = 0.42, moderate; AC1 = 0.83, excellent; *p* < 0.001). *Conclusions:* The 2017 ACC/AHA HTN guideline reported high agreement and detected more participants as hypertensive when compared with JNC 7 guideline for Bangladeshi postmenopausal women that demands further large-scale study in general population to clarify the current findings more precisely.

## 1. Introduction

Hypertension (HTN) is a major and most prevalent intermediate risk factor that accounts for 53% of global cardiovascular deaths annually [1,2]. Although this risk factor is highly preventable, its detection by qualified health professional is sub-optimal, only 53% in Bangladesh [3]. In this country, most of the clinicians and researchers are aligned with the application of “Seventh Report of Joint National Committee on Prevention, Detection, Evaluation, and Treatment of High Blood Pressure (JNC7)” [4] guidelines to detect HTN in various settings. As a result, most of the previous studies reported their HTN prevalence based on the JNC 7 guidelines. A national survey of Bangladesh detected 21% of the adult population as hypertensive which was more than the previous report [5,6]. However, well reported evidence on postmenopausal HTN among Bangladeshi population was inconsistent that demands further research on this issue.

It has documented that the numbers of postmenopausal women will increase from 467 million in 1990 to 1200 million in 2030 due to gradual increase of life expectancy among world population [7]. Although, aging is associated with increase of blood pressure both in men and women, the prevalence of HTN is greater in women than that in men [8,9] and 75% of women above 60 years of age becoming hypertensive in United States [9]. In this regard, a comprehensive review on postmenopausal HTN reported that health seeking behaviour, frequency of blood pressure measurement and even treatment adherence is better in women compared to men. However, women appear to be less controlled their HTN than their counterpart. So, there may be different mechanism responsible for postmenopausal HTN that needs separate treatment strategy to control the blood pressure [10]. A systematic review and meta-analysis on prevalence of HTN among the member states of South Asian Association for Regional Cooperation (SAARC) found that 8 studies out of 19 reported (3 from India, 1 from Bangladesh, 2 from Sri Lanka, 1 from Pakistan, and 1 from Maldives) the prevalence of HTN was higher among women than men and average prevalence showed no significant urban-rural difference [11]. As per our knowledge there is only one well reported study conducted in Bangladesh to detect postmenopausal HTN in a rural setting and it documented that the prevalence of postmenopausal HTN in this country was 49.2%, using JNC 7 cut-off values [12]. This prevalence was more than twice of the nationally representative prevalence of HTN (21%) among adults [13]. Thus, postmenopausal HTN is becoming gradually important for the developing countries like Bangladesh.

The American College of Cardiology and the American Heart Association (ACC/AHA) published updated guidelines in 2017 titled, The ACC/AHA Guidelines for the Prevention, Detection, Evaluation and Management of High Blood Pressure in Adults or the 2017 Hypertension Clinical Practice Guidelines [14]. This guideline defined HTN with new cut-off values (systolic blood pressure ≥ 130 mmHg, or diastolic blood pressure ≥ 80 mmHg) that have already changed the prevalence of HTN among different population. In Bangladesh [15] and Nepal [16], application of ACC/AHA guidelines has already documented the double of HTN prevalence among adult population. But it has claimed that the healthcare system of the developing countries has not yet ready to handle this huge burden of HTN and it will not be cost-effective as anticipated [17]. Again, the American College of Physicians and other guideline making organizations like National Institute for Health and Care Excellence (NICE) or European Society of Cardiology (ESC) have not yet accepted the guidelines [18]. Considering all of these controversies, the basic research question of this study was whether the cut-off values of these two guidelines statistically concordant at population level to detect different sub-types and staging of HTN. Hence this study aimed to evaluate the agreement between JNC 7 and 2017 ACC/AHA guidelines to detect postmenopausal HTN at different cut-off values and assess the change of prevalence followed by comparison between the guidelines.

## 2. Materials and Methods

### 2.1. Study Design

This cross-sectional study was conducted in a rural primary health care centre of Bangladesh among 265 postmenopausal women of 40–70 years of age. The sample size was determined using a cardiovascular disease (CVD) risk prevalence obtained from a study conducted among postmenopausal women of Nigeria, another developing country [19]. We selected the centre purposively and followed convenient sampling technique to collect the data. The subjects were screened as having no CVD based on their self-reported statement, clinical history and medical records review. Their postmenopausal status was defined as no menstrual bleeding for a period of at least 12 months and no other clinical condition causing amenorrhea [20]. Exclusion criteria included acute illness of the subjects or unwilling to participate or those with mental instability.

### 2.2. Data Collection and Measurement

The data collection procedure comprised of three phases. At first phase, face-to-face interview was conducted using pre-tested questionnaire adapted from STEP-wise approach to Surveillance (STEPS) of Noncommunicable diseases risk factors of World Health Organization (WHO) with appropriate modifications [21]. The questionnaire collected sociodemographic, reproductive and behavioral risk factors (tobacco use, alcohol intake, physical inactivity) related information of the participants. Their physical activity level was measured by Estimated Energy Requirement (EER) equation of the Dietary Reference Intakes (DRIs) Committee [22]. In second phase, anthropometric and blood pressure (BP) measurements were carried out following the methods described in ‘Noncommunicable disease risk factors survey Bangladesh 2010’ [23]. The anthropometric measurements included height, weight, waist circumference, hip circumference, and their derivatives like body mass index (BMI) and waist-hip ratio. These parameters were measured by a trained female assistant maintaining adequate privacy. In third phase, to measure biochemical parameters (blood glucose and blood lipids), 5 cc of venous blood was collected with aseptic precaution and again 3 cc was collected after two hours of 75-gm glucose intake. Oral glucose tolerance test (OGTT) was carried out for those who didn’t know their glycaemic status. Lipid profile including total cholesterol (TC), high-density lipoprotein cholesterol (HDL-C) and triglyceride (TG) was also measured at fasting state. The low-density lipoprotein cholesterol (LDL-C) was calculated according to the Friedewald formula using TC, HDL-C and TG [24]. To assure quality of measurements, we followed and maintained three criteria’s (1) use of standard measurement protocol (2) training by an experience trainer (3) application of robust equipment and (4) pre-testing to evaluate the whole procedure.

### 2.3. Blood Pressure Measurement and Definition of HTN

BP measurement procedures have been described previously [23]. In short, BP was measured by a trained physician using aneroid sphygmomanometer on the right arm in sitting position and hand in resting on handle of the chair or some objects. Following rested for at least 15 min the 1st reading was taken and a second reading was taken after 3 min resting interval. Systolic and diastolic measurement was taken by means of mm of Hg. The mean of the two measurements were used to determine the final value of blood pressure. To determine the prevalence of HTN and compare the findings, we applied both the cut-off values: ACC/AHA guidelines (systolic blood pressure ≥ 130 mmHg, or diastolic blood pressure ≥ 80 mmHg) [14] and JNC 7 guidelines (systolic blood pressure ≥ 140 mmHg, or diastolic blood pressure ≥ 90 mmHg) [4]. Again, those who reported previous diagnosis of HTN and/ use of anti-hypertensive drug were also diagnosed as hypertensive and included in the combined definition of HTN.

### 2.4. Ethical Consideration

The purpose of the study, necessity of invasive procedures and data safety issues were explained to the participants. All subjects gave their informed written consent for inclusion before they participated in the study. The study was conducted in accordance with the Declaration of Helsinki, and the protocol was approved by the Ethical Review Committee of Bangladesh University of Health Sciences (Identification number: BUHS/ERC/EC/16/024 (1/1)) on 28th January 2016. Reports of the blood test were delivered to the participants after 3 days. If any life-threatening condition detected in a participant, she was referred to a secondary or tertiary heath care centre to treat the condition.

### 2.5. Data Processing and Analysis

Data were analyzed using the Statistical Package for Social Science (SPSS) version 20.0 for Windows (SPSS, Inc., Chicago, IL, USA). All estimates of agreement were presented at 95% confidence interval in the tables. The JNC 7 classification has two stages (stage 1 and stage 2) and AHA/ACC has three stages (stage 1, stage 2 and hypertensive crisis). For this reason, we merged the last one (hypertensive crisis) with the stage 2 HTN to calculate the prevalence and agreement. Agreement between the 2017 ACC/AHA and JNC 7 guidelines was evaluated using Cohen’s Kappa (κ), Prevalence-Adjusted Bias-Adjusted Kappa (PABAK) and First-order Agreement Coefficient (AC1). To evaluate agreement, we considered JNC 7 guideline as gold standard and against it the new 2017 ACC/AHA guideline was examined. Here the overall agreement rate between two guidelines is the ratio of the number of hypertensive cases on which two raters are in agreement to the total number of hypertensive cases in the analysis. However, some agreement may happen by chance because of at least one guidelines’ guess or randomly rate a hypertensive case. Hence, we used Cohen’s Kappa to correct the amount of agreement due to chance. In addition to Cohen’s Kappa, we used PABAK and AC1 as Kappa is highly influenced by prevalence and bias of the two guidelines.

To overcome the limitations of Kappa and more justification of agreement among the guidelines, we also reported bias index (BI) and prevalence index (PI). Here PABAK adjusted the imbalances caused by differences in the prevalence and bias. On the other hand, we used AC1 to overcome the phenomenon which is well-known as kappa paradoxes (low Kappa at high agreement and high kappa at unbalance marginal distribution) and AC1 is not affected by bias at all [25,26]. This low kappa value may mislead the results in presence of kappa paradox. Hence, researchers suggested reporting additional values in addition to kappa to provide a clear picture of agreement which was another justification why we used PABAK, AC1, BI and PI [27]. The value of BI ranges from 0–1, with 0 indicating no bias and 1 implying that one guideline never identifies the condition while the other guideline always does. The PI also ranges from 0–1, with 0 indicating the prevalence of the condition is 50%, while 1 suggesting the prevalence of the condition is 0 or 100% [25,26]. In this study the κ value was considered statistically significant at a threshold of *p* < 0.05 and the levels of agreement were interpreted using Landis & Koch’s approach [28].

## 3. Results

### 3.1. Charactiristics of the Study Population

We recruited 265 postmenopausal women with mean age of 53.51 ± 7.5 years. Most of them were housewives (90.2%), illiterate (60.8%) and came from lower-middle income background (70.6%). Their mean age at menopause and duration of menopause was 44.83 ± 5.22 years and 8.79 ± 6.45 years respectively. About 16.2% respondents gave the previous history of HTN and/use of antihypertensive drugs. Among the risk factors of HTN, nearly three forth (73.2%) of the postmenopausal women were centrally obese and more than half (58.1%) were physically inactive. Again, nearly same proportion (45%) of participants used smokeless tobacco and practiced extra salt intake with meal. About 20% of the study population had diabetes and 25.7% had hypercholesterolaemia (Appendix A).

### 3.2. Prevalence of Postmenopausal HTN

The prevalence of postmenopausal HTN was 41.9% and 67.5% based on JNC 7 and 2017 ACC/AHA guideline respectively. Regarding staging of HTN, both the guidelines detected same proportion of respondents at stage 1 HTN (28.7%). Again, the prevalence of stage 2 HTN was more than five times higher as per 2017 ACC/AHA cut-off value than the JNC 7 (35.8% vs 6.8%). However, the prevalence of raised blood pressure (pre-hypertension/elevated) was ten times greater for JNC 7 guideline (32.5%) compared with the other guideline (3.8%). Both the guidelines categorized nearly same proportion of participants as normotensive, 32% for JNC 7 guideline and 31.7% for 2017 ACC/AHA guideline (Table 1).

### 3.3. Agreement between 2017 ACC/AHA and JNC 7 Guidelines

According to the subtypes of HTN, agreement was observed among 228 participants (86.03%) for isolated systolic hypertension (ISH), 170 participants (64.2%) for isolated diastolic hypertension (IDH) and 197 participants (74.3%) for combined systolic and/diastolic hypertension (SDH). As per Landis & Koch’s approach, substantial to excellent agreement (κ = 0.74, PABAK = 0.76, AC1 = 0.84; *p* < 0.001) for ISH, fair to moderate agreement for IDH (κ = 0.5, PABAK = 0.28, AC1 = 0.32; *p* < 0.001) and moderate agreement for combined SDH (κ = 0.51, PABAK = 0.48, AC1 = 0.49; *p <* 0.001) was detected respectively (Table 2).

Among the stages of hypertension, agreement was observed among 189 participants (71.3%) for pre-HTN/elevated BP, 113 participants (42.6%) for stage 1 HTN and 188 participants (70.3%) for stage 2 HTN. According to Landis & Koch’s approach, slight to excellent agreement (κ = 0.12, PABAK = 0.42, AC1 = 0.83; *p* < 0.001) was detected for pre-HTN/elevated BP, poor agreement (κ = −0.39, PABAK = −0.16, AC1 = −0.35; *p* < 0.001) was detected for stage 1 HTN and slight to substantial agreement (κ = 0.24, PABAK = 0.42, AC1 = 0.75; *p* < 0.001) was detected for stage 2 HTN (Table 2).

## 4. Discussion

Current study reported that the new 2017 ACC/AHA HTN guideline estimated more respondents as hypertensive than the old JNC 7 guideline and significant changes in prevalence was noted. Among the HTN sub-types and stages, highest agreement was observed for ISH and pre-HTN/elevated BP. For ISH both guidelines showed substantial to excellent agreement and for ‘pre-HTN/elevated BP’ they showed slight to excellent agreement. These findings to detect postmenopausal HTN for Bangladeshi adults aged 40–70 years was significant as data related to burden of postmenopausal HTN among Bangladeshi women is currently lacking. Moreover, agreement among the aforementioned HTN guidelines was not evaluated in such a low resource setting before.

The results of our study reported that the prevalence of postmenopausal HTN was 25.6% increased when ACC/AHA guideline implied instead of JNC 7 guideline. This finding was consistent with the other two nationally representative studies of Islam et al. [15] and Kibria et al*.* [29] where the prevalence differed 22.8% and 22.3% respectively for adult population of Bangladesh. The possible reasons for high prevalence of HTN among postmenopausal women of current study were estrogen deficiency and high risk factors burden especially central obesity, physical inactivity and high salt intake with meal. An animal model study also suggested that estrogen deficiency may lead to HTN in women [30]. The prevalence of HTN among 40–70 years age group of our study was 67.5% as per 2017 ACC/AHA guideline that showed similarity (prevalence 63%) with the recent study of Khera et al among US population of 40–75 years age group [31]. In that study the increased prevalence of HTN for US population according to 2017 ACC/AHA guideline was 26.8% which was also consistent with the findings of current study. The difference was that the aforementioned study [31] compared the new guideline with the eighth JNC HTN guideline instead of the JNC 7. In the same study, the new 2017 ACC/AHA guideline was also examined for an external population (China) to evaluate the impact and the findings was significantly differed in terms of increased prevalence (26.8% US vs 45.1% Chinese) and drug requirement (7 million US vs 55 million Chinese). This huge difference suggested assessing the international impact of new guideline, especially for developing countries where cost-effectiveness is a matter of public health policy implication. Although it has claimed that this increased prevalence will put extra burden for the health system of developing countries [16], evidence suggested that the reality is different [32]. This is because the fact is not the prevalence of HTN rather the difference in prevalence of patients who should receive antihypertensive therapy and importantly all hypertensive patients not necessarily eligible for medications as per guidelines. In this regard the study of Muntner et al. is important as they found the prevalence of adult population of United States (US) was increased 13.7% as per 2017 ACC/AHA guideline, but the prevalence of hypertensive population eligible for drug treatment was significantly low (1.9%) [33]. Thus, the cost-effectiveness ratio was not altered by the new guideline for US population, contradictory to the report of Khera et al. for Chinese population. Hence, only prevalence of HTN is not the indicator to evaluate the impact of 2017 ACC/AHA HTN guidelines for any health care system of developing countries like Bangladesh. Analysis on drug requirement as per new guideline has not yet reported by any study of Bangladesh that demands further initiative to continue the ongoing research that will help to remodel the existing national HTN guideline. This is because the studies of Khera et al. and Muntner et al. revealed that the drug requirement varied for non-US population and previously it was also reported that white people more adhere to anti-hypertensive therapy than the Asian population [32].

Among the classifications of HTN, this study detected high agreement for ISH and those at the stage of pre-HTN/elevated BP. Overall, HTN sub-types showed comparatively better agreement than the staging of 2017 ACC/AHA guideline. Among the sub-types and stages, both carry equal importance in management of HTN. However, stages are mostly useful in clinical decision making as management guidelines suggested treatment protocol based on stages. From public health point of view, both HTN sub-types and stages are important to prevent future complications. According to 2017 ACC/AHA cut-off value, here the ISH is defined as systolic blood pressure (SBP) of ≥ 130 mmHg and a diastolic blood pressure (DBP) of <80 mmHg. The elevated BP is defined as SBP of 120–129 mm Hg and a DBP of <80 mm Hg. High agreement of 2017 ACC/AHA guideline for ISH has various public health implications as it is considered as the major and strong cardiovascular risk factor [34]. It has evidenced that the ISH is the consequences of aging process and a major form of HTN among adults age 50 years and above [35]. This evidence is also in favour of our study as 75% of postmenopausal women in current study were 50 years and above. On the other hand, elevated BP is also associated with significant cardiovascular diseases (CVD) risk [36]. Here the elevated BP of 2017 ACC/AHA has overlapping with pre-HTN of JNC 7 guideline. The reason is the range of pre-HTN which is defined as SBP of 120–139 mmHg or DBP of 80–89 mmHg in JNC 7. The report of Marijana et al*.* informed this range of BP is a gray zone [32]. This is because different guidelines categorized this range of BP value in different ways, pre-HTN by JNC 7 and “high-normal BP” by European guidelines. The 2017 ACC/AHA defined the low range value of this gray zone as “elevated BP”. In our study, this elevated BP of 2017 AHA/ACC showed excellent agreement with the pre-HTN of JNC 7. From clinical point of view, this finding was important as individual with this range of BP will get more attention due to future risk of HTN and subsequent complications. The importance is also reflected in the findings of a meta-analysis included 16 prospective studies of Asia and other regions [36]. The meta-analysis reported that the high-range pre-HTN (SBP/DBP of 130–139/85–89 mmHg) was associated with 56% higher risk of CVD (relative risk (RR), 1.56; 95% CI, 1.36–1.78), whereas low-range pre-HTN (SBP/DBP of 120–129/80–84 mmHg) was associated with 24% higher CVD risk (RR, 1.24; 95% CI, 1.10–1.39).

In this study, we evaluated the agreement in the light of prevalence index and bias index. This is because kappa is highly dependent on prevalence and bias of the attribute under evaluation [25]. To overcome the kappa paradoxes, interpretation using other measures including PABAK and AC1 showed progressive improvement of agreement for ISH (κ = 0.74, CI: 0.66–0.82; PABAK = 0.76, CI: 0.68–0.84; AC1 = 0.84, CI: 0.76–0.92), pre-HTN/elevated BP(κ = 0.12, CI: −0.04–0.28; PABAK = 0.42, CI: 0.26–0.58; AC1 = 0.83, CI: 0.67–0.99) and stage 2 HTN(κ = 0.24, CI: 0.10–0.38; PABAK = 0.42, CI: 0.28–0.56; AC1 = 0.75, CI: 0.61–0.89). This improvement of agreement using PABAK and AC1 ensured authenticity of the findings of current study. Usually kappa paradox occurs when the observed proportion of agreement (*p*_0_) is high but the kappa value is low. This low kappa value is misleading in the presence of kappa paradox. In our study, kappa paradox was observed for pre-HTN/elevated BP and stage 2 HTN respectively. In both the cases, the observed agreement was high (pre-HTN/elevated BP, *p*_0_ = 71.3; stage 2 HTN, *p*_0_ = 70.9); however, the kappa value was low (pre-HTN/elevated BP, κ = 0.12; stage 2 HTN, κ = 0.24). Hence, it has suggested that if other measures are presented in addition to the obtained value of kappa, its use may be considered appropriate as it gives an indication of the likely impacts of prevalence and bias alongside the true value of kappa [37]. Thus from statistical background, reporting of agreement between2017 ACC/AHA and JNC 7 HTN guidelines using PI, BI, PABAK and AC1 was justifiable and made the results robust. Current study also reported moderate agreement for SDH (κ = 0.51, PABAK = 0.48, AC1 = 0.49; *p* < 0.001) which was better than the findings of IDH. This is because, the combined SDH showed consistent values without fluctuation from the range of moderate agreement. In this study, compared to other parameters of HTN, the stage 1 HTN showed poor agreement among the guidelines. The possibility of such finding might be due to large difference in the range of cut-off values of two guidelines (SBP of 130–139 mmHg or DBP of 80–89 mmHg for 2017 ACC/AHA compared to SBP of 140–159 mmHg or DBP of 90–99 mmHg for JNC 7).

Other than these statistical issues, from public health and clinical perspective, the ISH is more important than the IDH as it is associated with adverse cardiovascular and renal outcomes [38]. The Multiple Risk Factor Intervention Trial (MRFIT) reported that great majority of excess deaths occurred at borderline high SBP or stage 1 HTN that is the range of SBP from 130–159 mmHg [39]. The Systolic Blood Pressure Intervention Trial (SPRINT) also showed that treating to a target SBP of less than 120 mmHg reduced rates of high blood pressure complications by 25% and compared with the JNC 7 target cut-off value, lowered the risk of death by 27% [40]. To get these benefits of CVD risk reduction, attention should be given to the SBP at primary care setting. However, SBP is an underestimated CVD risk factors and control of it is more difficult than that of DBP. Hence HTN research suggested new treatment protocol or drugs for management of ISH [41]. Again, the 2017 ACC/AHA recommended starting treatment based on 10-years absolute CVD risk prediction and SBP is an important component of any CVD risk assessment tools. This is why we are more concerned about SBP rather DBP and our agreement in favour of ISH were significant. It is also important to mention that evidences on agreement between 2017 ACC/AHA guideline and JNC 7 guideline is currently lacking that makes it difficult to compare the findings of current study with others.

The present study had several limitations that should be considered. First of all, the cross-sectional design did not allow us to address any causal relationship. Second, small sample size limited the results to be generalized among other population. Third, we applied convenient sampling technique to collect the data that might be associated with selection bias. Explanation of these limitations is that the data of this study was based on a Master’s thesis in which CVD risk was estimated using different risk prediction tools and for that purpose sample size was calculated. The age limit was 40–70 years as it was the pre-requisite of the CVD risk prediction tools applied. As this was a self-funded Master’s thesis that had a defined/limited time period, we applied convenient sampling technique that enabled us to achieve the sample size we wanted in a relatively fast and inexpensive way.

Despite of these limitations, this study is important from the public health and clinical perspective. As per our knowledge, this is the first study of HTN that tried to evaluate the agreement between mostly used JNC 7 guideline and the emerging 2017 ACC/AHA HTN guideline. This is also the first study of Bangladesh that examined these guidelines to detect postmenopausal HTN in a rural primary healthcare setting. From gender perspective, this study is also important for Bangladesh as postmenopausal HTN is a neglected issue here and the policy makers merely address HTN among this specialized group of population. In addition to report the prevalence of HTN and agreement among the guidelines, this study also reported HTN related biochemical parameters and behavioral risk factors to inform the burden of contributing factors among the study population. Finally, simultaneous use of different indicators of agreement increased the strength of the results.

## 5. Conclusions

The 2017 ACC/AHA HTN guidelines showed high agreement with the JNC 7 guidelines that demands further large-scale studies to conduct in Bangladeshi population to establish its suitability. We recommend that these studies should be done in different health care settings and various age groups to generate evidence that will help to adopt the new 2017 ACC/AHA hypertension guideline for Bangladeshi population. To get the beneficial impact of new guideline, antihypertensive drug requirement also need to be estimated and compared with the existing national guideline to assess the cost-effectiveness and thereby remodeling of the primary health care system where appropriate.

## Figures and Tables

**Table 1 medicina-55-00315-t001:** Prevalence of hypertension according to its sub-types and staging among postmenopausal rural women of Bangladesh, *n* = 265.

Variables	*n* (%)
**Hypertension sub-types**	
JNC 7	
Isolated systolic hypertension	76 (28.7)
Isolated diastolic hypertension	57 (21.5)
Systolic and/diastolic hypertension *	111 (41.9)
2017 ACC/AHA	
Isolated systolic hypertension	113 (42.6)
Isolated diastolic hypertension	152 (57.4)
Systolic and/diastolic hypertension *	179 (67.5)
**Hypertension staging**	
JNC 7	
Pre-hypertension	86 (32.5)
Stage 1	76 (28.7)
Stage 2	18 (6.8)
2017 ACC/AHA	
Elevated blood pressure	10 (3.8)
Stage 1	76 (28.7)
Stage 2	95 (35.8)

JNC, Joint National Committee; ACC, American College of Cardiology; AHA, American Heart Association, * Included those on antihypertensive therapy and represented the total hypertensive subjects.

**Table 2 medicina-55-00315-t002:** Agreement of the JNC 7 and 2017 ACC/AHA guideline to detect hypertension among postmenopausal rural women of Bangladesh (*n* = 265).

HTN Sub-Types/Staging	JNC 7 Classification	ACC/AHA Classification		κ	BI	PI	PABAK	AC1
		Yes	No					
Isolated systolic HTN	Yes, *n*=76	76	0	0.74 (0.66–0.82)	−0.14 (−0.23 to −0.06)	−0.29 (−0.37 to −0.21)	0.76 (0.68–0.84)	0.84 (0.76–0.92)
	No, *n* = 189	37	152					
	Total, *n* = 265	113	152					
	Observed agreement, *p*_0_			86.03 (81.8–90.2)				
	*p*-value			<0.001				
Isolated diastolic HTN	Yes, *n* = 57	57	0	0.5 (0.35–0.65)	−0.36 (−0.46 to −0.26)	−0.21 (−0.31 to −0.11)	0.28 (0.18–0.38)	0.32 (0.22–0.42)
	No, *n* = 208	95	113					
	Total, *n* = 265	152	113					
	Observed agreement, *p*_0_			64.2 (58.4–70)				
	*p*-value			<0.001				
Combined systolic and/diastolic HTN	Yes, *n* = 111	111	0	0.51 (0.41–0.61)	−0.26 (−0.36 to −0.16)	−0.09 (−0.19–0.01)	0.48 (0.38–0.58)	0.49 (0.39–0.59)
	No, *n* = 154	68	86					
	Total, *n* = 265	179	86					
	Observed agreement, *p*_0_			74.3 (58.4–70)				
	*p*-value			<0.001				
Pre-HTN/elevated BP	Yes, *n* = 86	10	76	0.12 (−0.04–0.28)	0.29(0.13–0.45)	−0.64 (−0.80 to −0.48)	0.42(0.26–0.58)	0.83 (0.67–0.99)
	No, *n* = 179	0	179					
	Total, *n* = 265	10	255					
	Observed agreement, *p*_0_			71.3 (65.9–76.7)				
	*p*-value			<0.001				
Stage 1 HTN	Yes, *n* = 76	0	76	−0.39 (−0.49 to −0.29)	0	−0.43 (−0.53 to −0.33)	−0.16 (−0.26 to −0.06)	−0.35 (−0.45 to −0.25)
	No, *n* = 189	76	113					
	Total, *n* = 265	76	189					
	Observed agreement, *p*_0_			42.6 (36.6–48.6)				
	*p*-value			<0.001				
Stage 2 HTN	Yes, *n* = 18	18	0	0.24 (0.10–0.38)	−0.29 (−0.43 to −0.15)	−0.57 (−0.71 to −0.43)	0.42 (0.28–0.56)	0.75 (0.61–0.89)
	No, *n* = 247	77	170					
	Total, *n* = 265	95	170					
	Observed agreement, *p*_0_			70.9 (65.4–76.2)				
	*p*-value			<0.001				

All estimates of agreement are presented at 95% confidence interval, JNC, Joint National Committee; ACC, American College of Cardiology; AHA, American Heart Association; HTN, hypertension; BI, Bias index; PI, Prevalence index; PABAK, Prevalence-adjusted bias-adjusted kappa; AC1, First-order agreement coefficient

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
