# Peer review of "Agreement between 2017 ACC/AHA Hypertension Clinical Practice Guidelines and Seventh Report of the Joint National Committee Guidelines to Estimate Prevalence of Postmenopausal Hypertension in a Rural Area of Bangladesh: A Cross Sectional Study"

_medicina, 2019, doi:10.3390/medicina55070315_

Reviewer 1 Report

Re: Agreement between 2017 ACC/AHA Hypertension Clinical Practice Guidelines and Seventh Report of the Joint National Committee guidelines to estimate prevalence of postmenopausal hypertension in a rural area of Bangladesh

 The study examined the level of agreement between the two guidelines to estimate the prevalence of hypertension among postmenopausal women in a rural area of Bangladesh. The manuscript is clearly written, and methods and data presentation are adequate for the study aim. I only have minor comments.

 1.       Result: please report how many women (parentage) had a previous diagnosis of hypertension and/or use of anti-hypertensive drug in this study, thus were included in the combined definition of hypertension.

2.       Line 212: the prevalence of hypertension was 67.2% based on the 2017 ACC/AHA guideline (not 67.5%) according to data presented in Table 4.

Author Response

Response to Reviewer 1 Comments

Dear Sir, we would like to thank for your overall comment on our submitted manuscript. Here we have added our responses. The changes have marked by yellow background.

 Point 1: Result: please report how many women (parentage) had a previous diagnosis of hypertension and/or use of anti-hypertensive drug in this study, thus were included in the combined definition of hypertension.

 Response 1: We have reported it in the result section (line number 176)

 Point 2: Line 212: the prevalence of hypertension was 67.2% based on the 2017 ACC/AHA guideline (not 67.5%) according to data presented in Table 4.

 Response 2: There was an unintentional error in the cell of prevalence of hypertension in the Table 4 (now it is Table 1). We have corrected the prevalence in the table as 67.5% (67.2% was not accurate).

Reviewer 2 Report

The authors used the JNC 7 guideline as gold standard to evaluate the agreement on detecting hypertension with 2017 ACC/AHA guideline in postmenopausal women in rural area of Bangladesh. However, I have some comments on the results and conclusion part of this paper.

Major Comments:

1. The authors first reported the proportion of agreement observed, e.g., 86.03% for systolic hypertension, 64.2% for diastolic hypertension, then reported the Cohen’s Kappa (κ), PABAK, and AC1. The purpose of using Kappa et al is to correct the amount of agreement due to chance. This point should be indicated clearly in the method of “Data Processing and Analysis” part.

2. The results showed in the tables are not clear.

Taking systolic hypertension for example, the authors should use a 2×2 table to shows the categories of women diagnosed by two guidelines. The four cells (a, b, c, d) of the table are: number of women diagnosed as hypertension by both guideline (a), number of women diagnosed as hypertension by JNC 7 but not by ACC/AHA(b), number of women diagnosed as hypertension by  ACC/AHA nut not by JNC 7 (c), number of women diagnosed as non-hypertension by both guideline(d).

Based on the above 2×2 table, the authors can show the calculated results of proportion of agreement observed, Cohen’s Kappa (κ), PABAK, AC1, BI and PI.

Please show the results of diastolic hypertension, stage 1 hypertension, stage 2 hypertension, and elevated/pre-hypertension also as tables mentioned above.

The authors can consider combing the above tables as one table.

3. At present, the authors did not show the proportion of agreement observed, Cohen’s Kappa (κ), PABAK, and AC1 for stage 1 hypertension, stage 2 hypertension, and elevated/pre-hypertension, please add these results.

4. In the Method part, a big part is “2.4 Ascertainment of Hypertension Risk Factors”. I do not think it’s necessary to include this part, as these factors will not affect the comparison of the two guidelines (both guidelines used the same women population). Also, in “2.2. Data Collection and Measurement”, the authors have included information collection.

5. Table 4 should be the first Table of this manuscript, i.e., firstly report the prevalence of hypertension in each guideline, then compare the agreement between them. No need to include “Hypertension risk factors” in this table. The authors can describe percentages of these risk factors as population characteristics in the result part.

6. The authors mentioned “it indicated that the new 2017 ACC/AHA guideline has the potentiality to use suitably and interchangeably with the existing JNC 7 hypertension guideline”(lines 252-253), and “implementation of the new 2017 ACC/AHA guidelines may improve prevention effort to reduce hypertension related complications among postmenopausal women of Bangladesh.” The nature of the study and results of this paper cannot draw such conclusions, please delete or revise these descriptions.

Minor Comments:

1. Line 28, “In between”, delete in.

2. Line 51, delete “In accordance with this statement”.

3. Line 53, change “evidences” into “evidence”, “are” to “was”.

4. Line 58, “than that in men”.

5. Line 90, delete “”levels of”.

6. Please use short sentences rather than long sentences.

Author Response

Response to Reviewer 2 Comments

 Dear Sir,

 Thank you for your valuable feedback. According to your comments we have re-analyzed, revised, modified and rewrite the manuscript where applicable. Again we think, in our 1st submission we failed to address and elaborate some important issues that revealed lack of clarity. So in this version of submission we have added new information as per your suggestion. We believe it will improve quality of the manuscript. This is also important to mention that our corrections or changes have marked by yellow background.

 Major Comments:

 Point 1: The authors first reported the proportion of agreement observed, e.g., 86.03% for systolic hypertension, 64.2% for diastolic hypertension, then reported the Cohen’s Kappa (κ), PABAK, and AC1. The purpose of using Kappa et al is to correct the amount of agreement due to chance. This point should be indicated clearly in the method of “Data Processing and Analysis” part.

 Response 1: We have added the point in our “Data Processing and Analysis” part (line number 151-155).

 Point 2: The results showed in the tables are not clear.

 Taking systolic hypertension for example, the authors should use a 2×2 table to shows the categories of women diagnosed by two guidelines. The four cells (a, b, c, d) of the table are: number of women diagnosed as hypertension by both guideline (a), number of women diagnosed as hypertension by JNC 7 but not by ACC/AHA (b), number of women diagnosed as hypertension by ACC/AHA nut not by JNC 7 (c), number of women diagnosed as non-hypertension by both guideline (d).

 Based on the above 2×2 table, the authors can show the calculated results of proportion of agreement observed, Cohen’s Kappa (κ), PABAK, AC1, BI and PI.

 Please show the results of diastolic hypertension, stage 1 hypertension, stage 2 hypertension, and elevated/pre-hypertension also as tables mentioned above.

The authors can consider combing the above tables as one table.

 Response 2: According to your suggestion, we have re-analyzed and tabulated the findings in a new format. The observed agreement has added in the table and presented at 95% confidence interval.

 In our 2nd submission, we have added three new tables for pre-hypertension/elevated blood pressure (Table 5), stage 1 hypertension (Table 6) and stage 2 hypertension (Table 7) according to your suggested format.

 We think separate presentation of each table will be more beneficial for the readers.

 Point 3: At present, the authors did not show the proportion of agreement observed, Cohen’s Kappa (κ), PABAK, and AC1 for stage 1 hypertension, stage 2 hypertension, and elevated/pre-hypertension, please add these results.

 Response 3: We have added these results in the section “3.3. Agreement between 2017 ACC/AHA and JNC 7 Guidelines” (line number 199-206).

 Point 4: In the Method part, a big part is “2.4 Ascertainment of Hypertension Risk Factors”. I do not think it’s necessary to include this part, as these factors will not affect the comparison of the two guidelines (both guidelines used the same women population). Also, in “2.2. Data Collection and Measurement”, the authors have included information collection.

 Response 4: We have removed the part “2.4 Ascertainment of Hypertension Risk Factors” from the manuscript.

 Point 5: Table 4 should be the first Table of this manuscript, i.e., firstly report the prevalence of hypertension in each guideline, then compare the agreement between them. No need to include “Hypertension risk factors” in this table. The authors can describe percentages of these risk factors as population characteristics in the result part.

 Response 5: In our 2nd submission, we have presented the Table 4 as first table and risk factors have removed from this table. According to your suggestion, we re-arranged our results and discussion, i.e., firstly reported the prevalence of hypertension in each guideline, and then compared the agreement between them. We have added a new paragraph at the beginning of result section titled “3.1 Characteristics of study population” in which we reported the proportion of different risk factors.

 Point 6: The authors mentioned “it indicated that the new 2017 ACC/AHA guideline has the potentiality to use suitably and interchangeably with the existing JNC 7 hypertension guideline”(lines 252-253), and “implementation of the new 2017 ACC/AHA guidelines may improve prevention effort to reduce hypertension related complications among postmenopausal women of Bangladesh.” The nature of the study and results of this paper cannot draw such conclusions, please delete or revise these descriptions.

 Response 6: We have deleted the sentences.

 Minor Comments:

 Point 1: Line 28, “In between”, delete in.

 Response 1: We have deleted “in”

 Point 2: Line 51, delete “In accordance with this statement”.

 Response 2: We have deleted it.

 Point 3: Line 53, change “evidences” into “evidence”, “are” to “was”

 Response 3: We have changed “evidences” into “evidence” and “are” to “was”

 Point 4: Line 58, “than that in men”.

 Response 4: We have changed accordingly.

 Point 4: Line 90, delete “levels of”

 Response 5: We have deleted accordingly.

 Point 4: Please use short sentences rather than long sentences.

 Response 5: We have tried our best to take care of this issue.

 Special Note from the Authors

 We rechecked all data presented in the tables in our 1st submission. We found unintentional errors in coding of the variables namely pre-hypertension/elevated blood pressure, stage 1 and stage 2 hypertension. Due to miscoding, the prevalence of mentioned variables were either overestimated or underestimated in the Table 1 (present) in our 1st submission. In our 2nd submission, we have presented the corrected prevalence of these variables in the Table 1 and marked by yellow background. We want to resubmit our corrected data set if the respected Editor and reviewers consider resubmitting it.

Round  2

Reviewer 2 Report

I have no further comments. Just please the authors to consider combing Table 2—Table 7, they are all in the same format. 

Author Response

Dear Sir,

 Thank you for your valuable feedback. According to your comments we have brought changes where necessary. This is also important to mention that our corrections or changes have marked by yellow background.

 Comments:

 Point 1: I have no further comments. Just please the authors to consider combing Table 2—Table 7, they are all in the same format. 

 Response 1: We have combined Table 2-7 into a new Table 2. Accordingly we changed the table number in the results and discussion.